# Interleukin-6 as a Predictive Factor of Pathological Response to FLOT Regimen Systemic Treatment in Locally Advanced Gastroesophageal Junction or Gastric Cancer Patients

**DOI:** 10.3390/cancers16040757

**Published:** 2024-02-12

**Authors:** Katarzyna Marcisz-Grzanka, Beata Kotowicz, Aleksandra Nowak, Mariola Winiarek, Malgorzata Fuksiewicz, Maria Kowalska, Andrzej Tysarowski, Tomasz Olesinski, Jakub Palucki, Urszula Sulkowska, Agnieszka Kolasinska-Cwikla, Lucjan Stanislaw Wyrwicz

**Affiliations:** 1Department of Clinical Oncology and Radiotherapy, Maria Sklodowska-Curie National Research Institute of Oncology (MSCNRIO), Wawelska 15, 02-034 Warsaw, Poland; mariola.winiarek@nio.gov.pl (M.W.); agnieszka.kolasinska-cwikla@nio.gov.pl (A.K.-C.); 2Cancer Biomarker and Cytokines Laboratory Unit, Maria Sklodowska-Curie National Research Institute of Oncology (MSCNRIO), W.K. Roentgena 5, 02-781 Warsaw, Poland; beata.kotowicz@nio.gov.pl (B.K.); malgorzata.fuksiewicz@nio.gov.pl (M.F.); maria.kowalska@nio.gov.pl (M.K.); 3Department of Molecular and Translational Oncology, Maria Sklodowska-Curie National Research Institute of Oncology (MSCNRIO), W.K. Roentgena 5, 02-781 Warsaw, Poland; aleksandra.nowak2@nio.gov.pl (A.N.); andrzej.tysarowski@nio.gov.pl (A.T.); 4Department of Oncological Surgery and Neuroendocrine Tumors, Maria Sklodowska-Curie National Research Institute of Oncology (MSCNRIO), W.K. Roentgena 5, 02-781 Warsaw, Poland; tomasz.olesinski@nio.gov.pl; 5Department of Radiology, Maria Sklodowska-Curie National Research Institute of Oncology (MSCNRIO), W.K. Roentgena 5, 02-781 Warsaw, Poland; jakub.palucki@nio.gov.pl; 6National Cancer Registry, Maria Sklodowska-Curie National Research Institute of Oncology (MSCNRIO), Wawelska 15B, 02-034 Warsaw, Poland; urszula.sulkowska@nio.gov.pl

**Keywords:** gastric cancer, gastroesophageal junction cancer, IL-6, FLOT regimen, predictive biomarker

## Abstract

**Simple Summary:**

The response rate after neoadjuvant chemotherapy (NAC) remains limited. Moreover, there are currently no biomarkers enabling an individual prediction of therapeutic efficacy. The aim of this study was the identification of serum biomarkers of early response to NAC. The elevated level of IL-6 prior to treatment and cycle 2 of the FLOT regimen might be a predictor of pathological response to NAC in locally advanced gastric cancer (GC) or gastroesophageal junction (GEJ) cancer. The results were obtained from a small group of patients and currently cannot be used in everyday clinical practice. Confirmation of the results in a larger group of patients seems to be essential from a clinical point of view, bearing in mind that IL-6 plays a significant role in gastric cancer biology, particularly in metastasis formation and in the mechanism of chemotherapeutic resistance.

**Abstract:**

Background: Perioperative treatment is a gold standard in locally advanced gastric cancer or GEJ cancer in the Western population. Unfortunately, the response rate after neoadjuvant chemotherapy (NAC) remains limited. Moreover, there are currently no biomarkers enabling an individual prediction of therapeutic efficacy. The aim of this study was the identification of serum biomarkers of early response to NAC. Methods: We conducted this prospective study in the MSCNRIO in Warsaw, Poland. A total of 71 patients and 15 healthy volunteers gave informed consent. Complete blood count, carcinoembryonic antigen (CEA), carcinoma antigen 125 (CA125), carcinoma antigen 19.9 (CA19.9), and fibrinogen (F) were measured at baseline and before every cycle. Circulating tumour cells (CTCs) and interleukin-1β (IL-1β), interleukin-6 (IL-6), interleukin-8 (IL-8), and interleukin-10 (IL-10) were measured in a pilot group of 40 patients at baseline and before cycle two (C2) and cycle three (C3). Results: Of all the measured parameters, only the IL-6 serum level was statistically significant. The IL-6 level before C2 of chemotherapy was significantly decreased in the complete pathological response (pCR) vs. the non-pCR group (3.71 pg/mL vs. 7.63 pg/mL, *p* = 0.004). In all patients with an IL-6 level below 5.0 pg/mL in C2, tumour regression TRG1a/1b according to the Becker classification and ypN0 were detected in postoperative histopathological specimens. The IL-6 level before C1 of chemotherapy was significantly elevated in ypN+ vs. ypN0 (7.69 pg/mL vs. 2.89 pg/mL, *p* = 0.022). Conclusions: The trial showed that an elevated level of IL-6 prior to treatment and C2 might be a predictor of pathological response to NAC.

## 1. Introduction

Gastric cancer is a major problem influencing life expectancy due to its aggressive nature [1]. It is associated with a poor prognosis dependent on the tumour stage at presentation [2]. According to GLOBOCAN 2020, gastric cancer is the fifth most common cancer and the fourth leading cause of mortality worldwide [3]. The outcome of GC and GEJ adenocarcinoma following curative resection alone is predominantly dismal, which points to the necessity of the application of perioperative chemotherapy [4,5]. This approach in Western patients with locally advanced primary resectable GC and GEJ cancer has been a gold standard since both the MAGIC trial and the French FNCLCC/FFCD 97033 study [4,5]. The most effective type of chemotherapy, as reported by Al-Batran S. et al. in The Lancet in 2019, is the FLOT regimen, which induces more tumour responses than other regimens and improves the margin-free resection rate. FLOT is superior to ECF/ECX with respect to the complete pathological response (15% vs. 6%) and median overall survival (50 months vs. 35 months) [6]. As approximately 10–15% of patients fail to respond to this treatment, it is vital to conduct studies on the application of biomarkers in GC and GEJ cancer [6]. There are currently no biomarkers enabling the prediction of therapeutic efficacy and real-time tumour dynamics or the identification of patients at an increased risk of a poor pathological response. In non-responding patients to neoadjuvant chemotherapy, such markers could allow clinicians to apply a more individualised approach, e.g., avoiding exposition to the potential toxicity of unnecessary chemotherapy, thus improving the quality of life and making it possible to perform earlier surgery as well as reduce the cost of treatment.

Scientific research over the last decade has explained the fundamental molecular mechanisms and provided conclusive evidence that inflammation is now established as a hallmark of cancer. The tumour microenvironment, which includes inflammatory cells or inflammatory mediators such as cytokines, chemokines, growth factors, prostaglandins, and stromal activation, plays a decisive role at various stages of tumour development, including initiation, malignant transformation, promotion, and invasion, as well as metastasis formation [7,8,9]. Pro-inflammatory cytokines (IL-1, IL-6, IL-8, TNF-α) are responsible for metastasis promotion and cachexia, while anti-inflammatory immunosuppressive cytokines such as in IL-10 are reported to be a marker of a higher stage of the disease [10,11]. Interleukin-6 (IL-6) and interleukin-8 (IL-8) are important pro-angiogenic factors in gastric cancer through the induction of vascular endothelial growth factor (VEGF) [12,13]. Both IL-1 and IL-6 are involved in the growth of neoplastic cells in gastric cancer and the metastasis formation [14,15]. In addition to pro-inflammatory cytokines, activation of the coagulation and angiogenesis systems is believed to be associated with the development of cancer [16]. Activated platelets are the source of VEGF, which is responsible for the promotion of neoangiogenesis. Lymphocytes participate in both humoral and cellular anti-tumour immune response [17].

In numerous studies on patients with various types of cancer, the lymphocyte-to-monocyte ratio (LMR), neutrophil-to-lymphocyte ratio (NLR), and platelet-to-lymphocyte ratio (PLR) were assessed as prognostic markers for overall survival (OS), disease-free survival (DFS), and progression-free survival (PFS) [17,18,19]. The study by Lian L. et al. conducted on patients with primary operable gastric cancer showed that preoperative low levels of PLR and NLR were correlated with better clinicopathological features, including a lower depth of tumour invasion, fewer lymph node metastases, and an early-stage cancer based on the TNM classification according to the AJCC [20].

Additionally, low levels of leukocytes and lymphocytes prior to systemic adjuvant therapy were a predictor of poor outcome in response to this treatment [21]. Patients with primary metastatic gastric cancer undergoing palliative systemic treatment who had a low NLR level before treatment had a statistically significantly better disease control rate (DCR), longer progression-free survival (PFS), and longer overall survival compared to patients with an initially high NLR level [21,22]. Arigami T. et al. developed a new scoring system (F-NLR) based on fibrinogen concentration (F) and the NLR ratio as a predictive and prognostic factor to chemotherapy or chemoradiotherapy in patients with advanced gastric cancer. Higher F-NLR values were significantly more frequent in the subgroup of patients with disease progression during treatment [23].

In 1869, during the autopsy of a patient with metastatic cancer, Thomas Ashworth first observed that cells similar to those of the primary tumour were present in peripheral blood. These cells are circulating tumour cells (CTCs): rare cancer cells released from the tumour into the bloodstream, which are thought to play a key role in cancer metastasis. Many studies have shown the identification of CTCs in patients with various types of cancer and their usefulness as a marker of response to systemic treatment [24].

In view of the above considerations, we selected the parameters as potential predictive factors of neoadjuvant chemotherapy in patients with locally advanced GC and GEJ cancer.

## 2. Materials and Methods

We conducted this prospective study in the Maria Sklodowska-Curie National Research Institute of Oncology in Warsaw, Poland, in order to identify serum biomarkers of early response to NAC from collected biomaterial. The trial was performed in accordance with the principles of the Declaration of Helsinki, and the protocol was approved by the Local Bioethics Committee at the Maria Sklodowska-Curie National Research Institute of Oncology (MSCNRIO) in Warsaw (approval number 51/2016/2017). A total of 71 patients and 15 healthy volunteers gave their informed consent.

### 2.1. Inclusion and Exclusion Criteria

The main eligibility criteria included the following: written informed consent for participation in the trial, patients with histopathologically confirmed GC or GEJ adenocarcinoma of a clinical stage cT2-T4/cN-any or cT-any/cN+, ECOG (Eastern Cooperative Oncology Group) performance status < 2, adequate liver, kidney, and hematologic function, and age ≥ 18 years old. The main exclusion criteria were the following: evidence of distant metastasis, history of other primary malignancies, prior chemotherapy or radiotherapy, active or documented prior autoimmune or inflammatory disorder, current or prior use of immunosuppressive medication or corticosteroids exceeding 10 mg/day of prednisone or its equivalent, allergy to iodine contrast agent, concomitant disease (coronary heart disease, arrhythmia, stroke) preventing administration of chemotherapy according to protocol, pregnancy, and breastfeeding.

### 2.2. Patient Treatment and Procedure

Clinical stage at baseline was evaluated with oesophagogastroduodenoscopy (OGD), computed tomography (CT), or magnetic resonance imaging (MRI) scan of the chest, abdomen, and pelvis and physical examination. Diagnostic laparoscopy was not performed in any patient as the Polish standards of care state that it is recommended but not mandatory. We administered treatment according to the original protocol without modification. FLOT administration consisted of four preoperative and four postoperative cycles (during each 2-week cycle, we administered docetaxel 50 mg/m^2^ on day 1, oxaliplatin 85 mg/m^2^ on day 1, leucovorin 200 mg/m^2^ on day 1, and 5-FU 2600 mg/m^2^ as 24 h infusion on day 1). Patients were assessed according to their medical history, physical examination, weight, ECOG performance status, complete blood count, CEA, CA125, CA19.9, and fibrinogen at baseline and before the start of every cycle. CTCs and IL- 1β, IL-6, IL-8, and IL-10 were measured in a pilot group of 40 patients at baseline and before the start of C2 and C3. We graded adverse events according to CTCAE 5.0 (Common Terminology Criteria for Adverse Events v5.0, 27 November 2017) before each cycle, and we used granulocyte colony-stimulating factor (G-CSF) for primary prophylaxis of febrile neutropenia. Chemotherapy was continued according to protocol unless written informed consent was withdrawn, unacceptable toxicity occurred, or progression of the disease was observed. In order to confirm the absence of progression of disease or occurrence of metastases, a CT or MRI scan of the chest, abdomen, and pelvis was performed between 2 and 4 weeks following the completion of the last cycle of preoperative chemotherapy. Tumour response was determined according to the Response Evaluation Criteria in Solid Tumours 1.1 (RECIST v1.1). Surgery was scheduled for 4–6 weeks following the completion of the last cycle of chemotherapy. Pathological tumour regression (TRG) of the primary tumour to NAC was evaluated according to the Becker classification, which classifies pathological response as follows: TGR1a, no residual tumour/tumour bed; TGR1b, <10% residual tumour/tumour bed; TGR2, 10–50% residual tumour/tumour bed; and TGR3, >50% residual tumour/tumour bed.

### 2.3. Biochemical Analysis

Venous blood collection VACUETTE^®^ was performed using the VACUETTE^®^ system (Greiner, Kremsmünster, Austria). The Sysmex XN-550 haematology analyser (Sysmex, Kobe, Japan) was used for the analysis of differential white blood cell count following the manufacturer’s protocol. The lymphocyte-to-monocyte ratio (LMR) was calculated by dividing an absolute count of lymphocytes (10^9^/L) by an absolute count of monocytes (10^9^/L). The platelet-to-lymphocyte ratio (PLR) was calculated by dividing an absolute count of platelets (10^9^/L) by an absolute count of lymphocytes (10^9^/L), and derived NLR (dNLR) was calculated by dividing an absolute count of neutrophils (10^9^/L) by an absolute leukocyte number (10^9^/L) minus absolute neutrophil number (10^9^/L) (neutrophil absolute number/(leucocyte absolute number—neutrophil absolute number)). The neutrophil-to-lymphocyte ratio (NLR) was calculated by dividing an absolute count of neutrophils (10^9^/L) by an absolute count of lymphocytes (10^9^/L). Plasma fibrinogen (F) was determined from blood plasma collected on sodium edetate (EDTA) using the Clauss method with Fibrinogen-C XL reagent in the ACL TOP 500 (Werfen, Barcelona, Spain) coagulation analyser according to the manufacturer’s recommendations. F-NLR score was based on plasma fibrinogen (F) and NLR. Patients with hyperfibrinogenaemia (>400 mg/dL) and high NLR (>3.0) received 2 points. Patients with only one of the above-mentioned abnormalities in biochemical parameters received 1 point, while those with a fibrinogen concentration <400 mg/dL and low NLR (<3.0) received 0 points. We established cut-off values for NLR and plasma fibrinogen concentration based on previously published data [22,23,25,26]. The tumour marker levels (CEA, CA125, CA19.9) were determined with electrochemiluminescence with Roche kits in the Cobas E601 system (Roche, Basel, Switzerland). The cut-off points for the markers were set according to the manufacturer’s recommendations. The serum concentrations of IL-1β, IL-6, IL-8, and IL-10 were determined by using an enzyme-linked immunosorbent assay (ELISA) with R&D Systems (Minneapolis, MN, USA) according to manufacturer’s recommendations.

### 2.4. Molecular Detection of Circulating Tumour Cells (CTCs)

Molecular detection of CTCs was performed by assessing the mRNA expression of tumour-associated markers (*CEA*, *CK19*, *survivin*). The VACUETTE^®^ system was used for venous blood collection. A 2.5 mL sample of peripheral venous blood from all of the patients and healthy volunteers was collected into PAXgene Blood RNA tubes (Qiagen, Hilden, Germany). Micro-centrifuge was used for purification and isolation of peripheral blood mononuclear cells (PBMCs). The RNA isolation was performed using the PAXgene Blood RNA Kit (Qiagen, Hilden, Germany) according to the manufacturer’s instructions and the QIAcube automatic nucleic acid isolation apparatus (Qiagen, Hilden, Germany). The amount of RNA was measured using a Quantus Fluorometer (Promega, Madison, WI, USA). The measurement was performed by using a fluorescent RNA-specific dye—QuantiFluor RNA System (Promega, Madison, WI, USA). The amount of RNA was expressed in ng/µL. A spectrophotometric test was performed in order to check the purity of the isolated RNA, with absorbance measured at 260 and 280 nm. The NanoDrop ND2000 device was used for the measurement (Thermo Fisher Scientific, Waltham, MA, USA). Then, on the basis of the ratio of A260 and A280, the instrument determined the degree of RNA purity. RNA was considered sufficiently purified material for further analysis if this ratio was approximately 2. Reverse transcription reactions were performed with the SuperScript IV VILO kit Master Mix with ezDNase enzyme (Thermo Fisher Scientific, Waltham, MA, USA). Measurement of the expression of reference genes (*TBP*, *HPRT*, *SDHA*, *YWHAZ*, *HPRT*, *GAPDH*, *ZNF410*) and marker genes (*CK19*, *CEA*, *survivin*) was performed using the real-time polymerase chain reaction method (real-time PCR, qPCR); it is presented in Appendix A (Table A1). Quantitative PCR reaction was performed using the ABI PRISM 7500 Applied Biosystems 7500 Fast Real-Time PCR System instrument (Applied Biosystems, Carlsbad, CA, USA). The reaction mixture consisted of TaqMan^®^ Gene Expression Master Mix 38 (Thermo Fisher Scientific, Waltham, MA, USA), TaqMan^®^ probes specific for selected genes (Thermo Fisher Scientific, Waltham, MA, USA), and cDNA matrix. Based on the qualitative assessment, three reference genes were selected and served as internal controls for further studies. The three selected genes were as follows: *TBP*, *HPRT*, and *ZNF410*. These genes were characterised by the highest stability of all the tested genes and showed constant expression in both patients and healthy volunteers. Where possible, quantitative analysis of the expression of *CK19* and *CEA* marker genes and survivin was performed. The expression value was calculated according to the comparative method. The value of the relative expression levels allowed us to estimate the changes in the expression of selected marker genes in patients with gastric cancer as compared to healthy volunteers.

### 2.5. Statistical Analysis

Statistical analysis was performed in R software (version 4.1.2). Age was described with median and range; the level of IL-6 was described with mean and standard deviation or median and interquartile range, depending on distribution normality. Categorical variables were presented as the absolute frequency and proportion of the group. Distribution normality was verified with the Shapiro–Wilk test, accompanied by skewness and kurtosis. Variance homogeneity was assessed with Levene’s test. Comparisons between prognosis groups were performed with *t*-Student test, Mann–Whitney U test, one-way ANOVA analysis, and Kruskal–Wallis test, as appropriate. Post hoc multiple comparison was conducted with the Dunn test with Bonferroni adjustment. Receiver operating characteristic (ROC) analysis was conducted in order to identify parameters with high potential to predict prognosis group. Optimal thresholds were calculated with Youden index.

## 3. Results

Between January 2018 and November 2019, a total of 71 patients gave their informed consent and started treatment. However, the final data analysis was conducted on 61 patients at the age of 30–77 (median 63 years; 52.5% male and 47.5% female). Two patients did not meet the inclusion criteria as they were not primary resectable, and we lost eight patients during preoperative treatment. The full preoperative treatment of four cycles of the FLOT regimen was administered to 93.4% (57) of the patients. CTCs and ILs were measured in a pilot group of 40 patients. The baseline characteristics of the patients, body mass index (BMI), and surgical and pathology results of the treatment are presented in Appendix A (Table A2, Table A3 and Table A4).

We did not find any statistical significance of CEA, CA19.9, CA125, IL-1β, IL-8, IL-10, LMR, NLR, dNLR, PLR, or CTCs in the complete pathologic response (pCR) vs. non-pCR group and ypN0 vs. ypN+ group, so we decided not to perform statistical analysis for the TRG–Becker subgroup. The data are presented in Appendix A (Table A5 and Table A6).

As NLR was not statistically significant in this subgroup, we did not take F-NLR un-der consideration in further analysis. Due to the fact that only eight patients had positive CTC-*CEA*, which was below ten cases, CTCs measured by the expression level for *CEA* were not taken into account in the statistical calculations.

Only the IL-6 serum level was found to be a potential biomarker of the pathological response to NAC. The IL-6 serum level before C2 of chemotherapy was significantly elevated in the non-pCR vs. the complete pathological response (pCR) group (7.63 pg/mL vs. 3.71 pg/mL, *p* = 0.004), see Figure 1.

The receiver operating characteristic (ROC) curve showed the predictive power of IL-6. The optimal threshold for diagnosing pCR was 5.0 pg/mL (AUC = 0.826, 95% CI: 0.698–0.954, *p* = 0.001), see Table 1 and Figure 2.

In all patients with an IL-6 serum level below 5.0 pg/mL in C2, tumour regression TRG1a/1b according to the Becker classification was detected in postoperative histopathological specimens. Due to the small sample size, the pCR group was defined as TGR-1a/1b and ypN0. A similar relationship was found in the ypN0 vs. the ypN+ group. The IL-6 serum level before C1 of chemotherapy was significantly elevated in ypN+ vs. ypN0 (7.69 pg/mL vs. 2.89 pg/mL, *p* = 0.022). The ROC curve showed the predictive power of IL-6. The optimal threshold for diagnosing ypN0 was 5.0 pg/mL (AUC = 0.751, 95% CI: 0.568–0.934, *p* = 0.017), see Table 2 and Figure 3.

A significant difference in the IL-6 serum level before C2 of chemotherapy was recognised comparing the TGR1, TGR2, and TGR3 groups (3.76 pg/mL vs. 7.07 pg/mL vs. 9.43 pg/mL, respectively, *p* = 0.016), see Figure 4.

Pairwise comparisons indicated TGR1 and TGR3 as the groups with a significant difference in the IL-6 serum level. The ROC analysis was performed to verify the predictive power of IL-6 for diagnosing TGR groups against each of the two other groups. Good diagnostic quality was identified when IL-6 was used to differentiate TGR1 from TGR2 and TGR1 from TGR3. The optimal threshold for diagnosing TGR1 vs. TGR2 was 5.16 pg/mL (AUC = 0.856, 95% CI: 0.674–1.000, *p* = 0.005). The optimal threshold for diagnosing TGR1 vs. TGR3 was 6.93 pg/mL (AUC = 0.796, 95% CI: 0.596–0.997, *p* = 0.004), see Table 3 and Figure 5.

The ROC analysis did not show a significant outcome of using IL-6 as a prognosis parameter for the TGR2 vs. TGR3 groups (*p* > 0.05), which means that IL-6 had good ability to predict that patients belonged to the TGR1 group, while its ability to distinguish the TGR2 group from the TGR3 group was not proved.

## 4. Discussion

Gastric cancer treatment no longer involves surgery alone but over the past decade has become a multimodality treatment. Most notably, the MAGIC trial and the French FNCLCC/FFCD 97033 trial demonstrated a significant survival benefit of perioperative treatment, which is currently a gold standard in the Western population [4,5,6]. The response to neoadjuvant chemotherapy constitutes a substantial prognostic factor for disease-free survival and overall survival [27,28,29]. As was demonstrated by the University of Texas MD Anderson Cancer Center, the ypStage provides reasonable survival prediction based on TNM grouping, whereas the clinical stage is not useful [30]. In patients with tumour downstaging, disease-free survival and overall survival are longer than in patients without response to preoperative chemotherapy, and the best outcome is observed in patients with pathological complete tumour regression [27,28,29]. Unfortunately, the response rate after NAC remains limited [4,5,6]. Moreover, there are currently no biomarkers enabling an individual prediction of therapeutic efficacy and real-time tumour dynamics or identifying patients at increased risk of a poor pathological response. In patients non-responsive to neoadjuvant chemotherapy, such markers could make it possible to determine the optimal balance between the risks and benefits of avoiding NAC in patients with locally advanced GC or GEJ cancer.

We know from previous analyses that both preoperative and postoperative serum levels of CEA, CA19.9, and CA125 are good prognostic factors in the surgical treatment of non-metastatic gastric cancer [31,32,33]. We did not demonstrate the usefulness of these markers as predictive biomarkers of early response to neoadjuvant treatment, probably due to the fact that more time is required to observe changes in the serum levels of these markers.

It was not possible to achieve statistical significance for LMR, NLR, dNLR, PLR, and CTCs as biomarkers, which are usually associated with the prognosis and response to treatment in advanced disease [20,21,22,34,35,36]. The cases included in this study were also less advanced: only 8% of the patients were cT4, and 26% of the patients were clinical stage III-A-C (TNM according to the AJCC—the 8th edition). This may also be the reason why we did not observe the usefulness of IL-10, which is mainly associated with a higher stage of the disease [11,37].

IL1β and IL-8 are important pro-angiogenic factors which participate in the growth of cancer cells, play a role in metastasis promotion, and have a pleiotropic effect on immune cells [12,14,38,39]. However, a better understanding of these mechanisms is needed in order to attempt to use them as markers of treatment response.

With regard to circulating cancer cells, we should bear in mind that they rarely occur in peripheral blood and that there are technical limitations of various assay methods. Currently, there is no well-established method available for determining CTCs in gastric cancer [40,41]. Based on previous data, we chose tumour-related mRNA (*CEA*, *CK19*, *survivin*) for the detection of circulating tumour cells in gastric cancer patients with the use of a reverse-transcriptase polymerase chain reaction (real-time PCR) method [42,43,44].

In our prospective, single-institution trial, we showed that out of all the measured parameters, only the elevated level of IL-6 prior to the start of treatment and C2 might be a predictor of pathological response to neoadjuvant chemotherapy. A significant difference in the IL-6 serum level before C2 of chemotherapy was recognised comparing the TGR1, TGR2, and TGR3 groups (3.76 pg/mL vs. 7.07 pg/mL vs. 9.43 pg/mL, respectively, *p* = 0.016). We had previously presented data suggesting that tumour regression grade after neoadjuvant treatment could be a prognostic factor in patients with locally advanced GC or GEJ cancer patients undergoing radical treatment. According to the Becker classification, 12% TRG1a (27), 14% TRG1b (31), 31% TRG2 (69), and 38% TRG3 (84) patients were reported, and the median overall survival (mOS) was 53.7 (95% CI; 48.0–59.3), 51.9 (95% CI; 45.9–57.9), 42.7 (95% CI; 36.2–49.2), and 28.3 (95% CI; 37.7–44.7), respectively (*p* < 0.001) [45]. This indicates a similarity with the results of other researchers (e.g., Athauda A. et al., Davarzani N. et al.) [29,46], although more investigation of this issue seems to be required.

Patients who did not experience pathological complete response (pCR) had statistically significantly higher serum levels in C2 than non-pCR patients. Similarly, node-positive (ypN+) patients had statistically significantly higher serum levels before the start of treatment than node-negative (ypN0) patients. Multivariate analysis by Smyth E. et al. demonstrated that the presence of lymph node metastases was the only factor independently predictive of overall survival in patients after NAC [27]. The latest data presented by Athauda A. et al. confirmed that lymph node status in the resection specimen is the single most important determiner of survival [29].

Our prospective pilot study is the first analysis of the utility of IL-6 as a predictive biomarker of early response to NAC.

IL-6 is a key immunomodulatory cytokine, which is involved in the orchestration of the innate and acquired immune system and plays an important role in the regulation of various homeostatic to pathological processes such as immune disease and cancers [47]. Studies by Kai H. et al. and Ito R. et al. show that IL-6 is involved in the growth of gastric cancer cells and the formation of metastases [14,15]. IL-6 is an important pro-angiogenic factor in gastric cancer through the induction of the VEGF [13]. A significant correlation was observed between the serum concentration of IL-6 and the tumour stage, depth of tumour invasion, lymphatic invasion, and venous invasion, as well as lymph node metastasis [37,48]. Increasing data suggest that IL-6 plays a crucial role in the modulation of the function and activity of tumour-associated immune cells [49]. IL-6 is a cancer-associated fibroblast (CAF)-specific secretory protein and a contributor to the dynamic crosstalk between tumour cells and the microenvironment, which is essential for tumour growth, invasion, and metastases. The epithelial–mesenchymal transition (EMT) of gastric cancer cells is induced by CAF-secreted IL-6. CAF-secreted IL-6 activates the Janus kinase (JAK) 1 signal transducer and activator of transcription 3 signal transduction (STAT) pathway in GC cell lines. The aberrantly hyperactivated IL-6/JAK/STAT3 pathway is generally associated with a poor clinical prognosis [50,51,52]. In vitro and in vivo studies showed that CAF-secreted IL-6 is a very important contributor of chemoresistance in GC. The interaction of CAFs with tumour cells may induce a more aggressive phenotype of cancer cells and confer 5-fluorouracyl resistance to gastric cancer cell lines through the inhibition of apoptosis [53]. This is extremely important as 5-fluorouracyl is the main cytostatic agent widely used in both perioperative and palliative treatment [4,5,6].

IL-6, apart from its role in tumourigenesis, has also been implicated in causing muscle wasting and caxechia. Caxechia is a significant clinical problem in gastric cancer patients, as well as a prognostic factor of the disease [54,55,56,57,58]. We are also in the process of preparing data on sarcopenia measured with the Hounsfield Unit Average Calculation (HUAC) of the psoas muscle and the Total Psoas Index (TPI) in CT performed prior to neoadjuvant chemotherapy and surgery. Subsequently, we aim at finding out whether there is a correlation of HUAC and TPI with inflammatory parameters.

In light of the above data, our study results seem to be of clinical importance. If the results are confirmed in a larger group of patients, the measurement of IL-6 serum level prior to start of treatment and prior to the administration of cycle 2 of neoadjuvant chemotherapy will enable the quick identification of ypN+ and non-pCR patients with a poor prognosis. If the effect of IL-6 on the induction of resistance to chemotherapy is also taken into consideration, it will be the basis for testing the efficacy of a combination of perioperative chemotherapy with IL-6 receptor inhibition [59]. Currently, there is an ongoing EMPOWER (NCT04333706) clinical trial of the combination of sarilumab (IL-6R inhibitor) plus capecitabine in triple-negative breast cancer patients in stage I-III with high-risk residual disease [60].

## 5. Conclusions

The above data suggest that IL-6 may be a predictive biomarker of pathologic response to neoadjuvant chemotherapy in patients with GC and GEJ cancer. The results were obtained in a small group of patients and currently cannot be used in everyday clinical practice. Confirmation of the results in a larger group of patients seems to be essential from a clinical point of view, bearing in mind that IL-6 plays a significant role in gastric cancer biology, particularly in metastasis formation and in the mechanism of chemotherapeutic resistance.

## Figures and Tables

**Figure 1 cancers-16-00757-f001:**
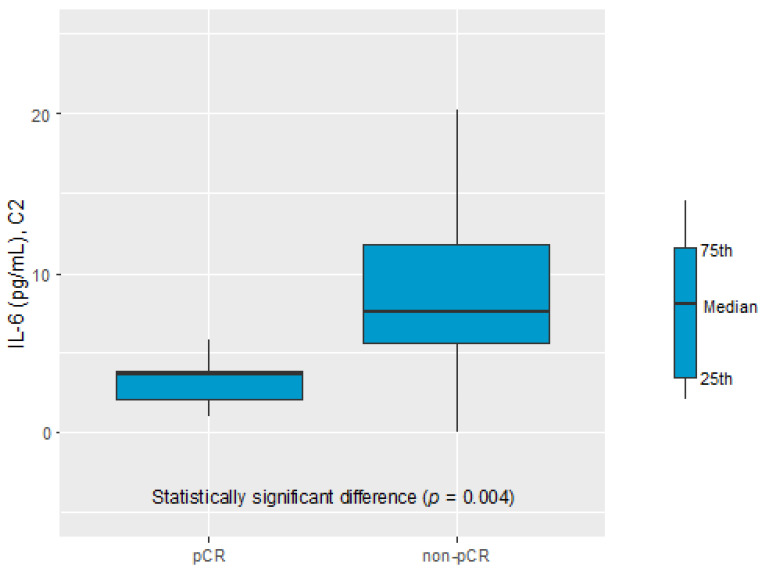
IL-6 in C2 in prognosis pCR and non-pCR group.

**Figure 2 cancers-16-00757-f002:**
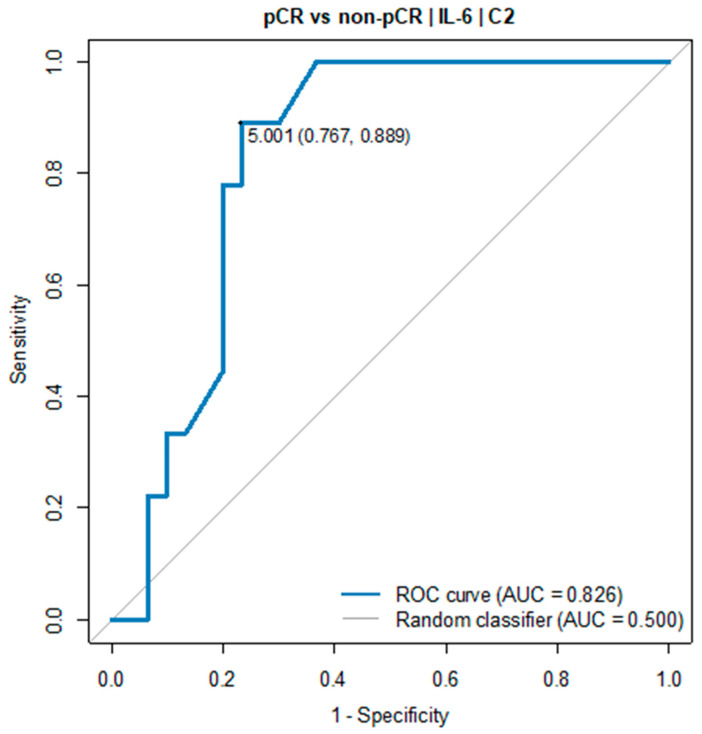
ROC curve for IL-6 (C2) as a diagnostic test between pCR and non-pCR group.

**Figure 3 cancers-16-00757-f003:**
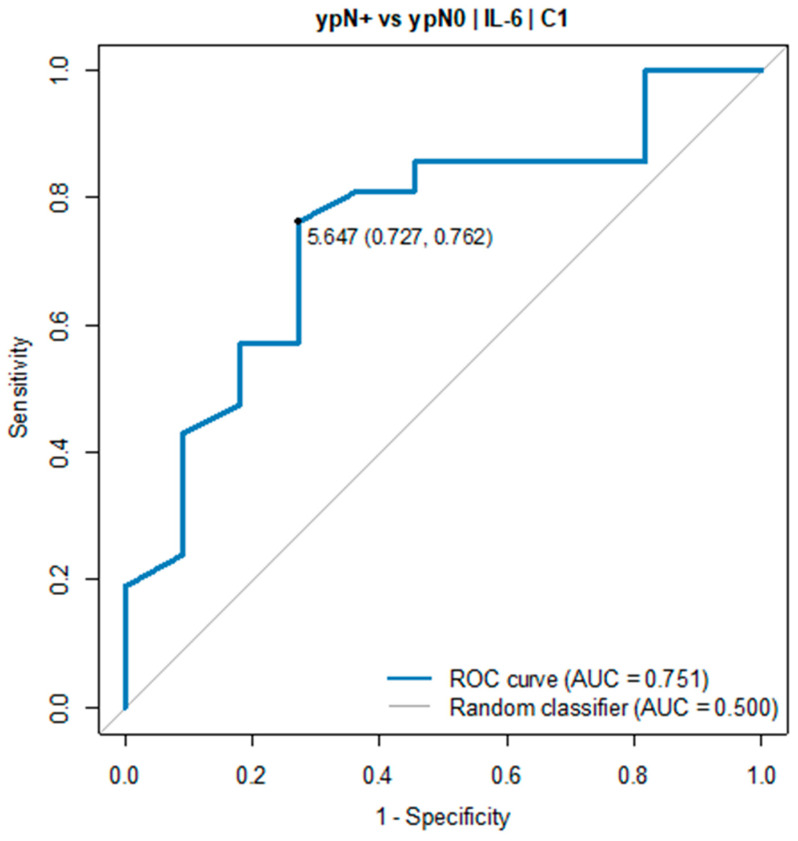
ROC curve for IL-6 (C1) as a diagnostic test between ypN+ and ypN0 group.

**Figure 4 cancers-16-00757-f004:**
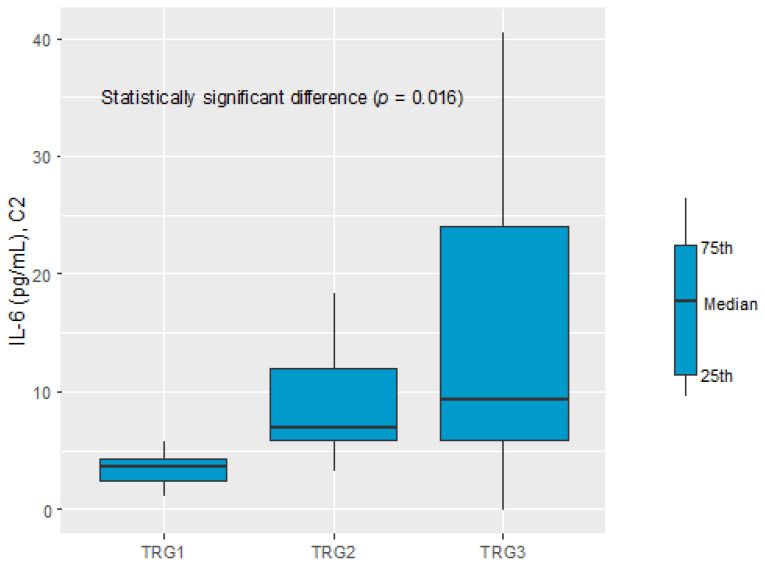
IL-6 in C2 within prognosis groups: TRG1, TRG2, and TRG3.

**Figure 5 cancers-16-00757-f005:**
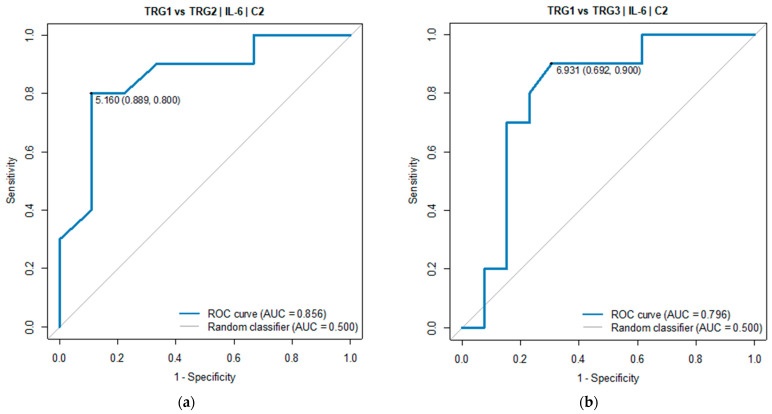
ROC curve for IL-6 (C2) as a diagnostic test between (**a**) TRG1 and TRG2 groups and between (**b**) TRG1 and TRG3 groups.

**Table 1 cancers-16-00757-t001:** Optimal thresholds for diagnosing pCR vs. non-pCR group ^1^.

	Optimal Threshold	AUC (95% CI)	Sensitivity	Specificity	Accuracy	PPV	NPV	*p*
Measurement C2								
IL-6 [pg/mL]	5.00	0.826 (0.698–0.954)	0.89	0.77	0.79	0.53	0.96	0.001

^1^ pCR group: *n* = 12; non-pCR group: *n* = 28.

**Table 2 cancers-16-00757-t002:** Optimal thresholds for diagnosing ypN+ vs. ypN0 group ^2^.

	Optimal Threshold	AUC (95% CI)	Sensitivity	Specificity	Accuracy	PPV	NPV	*p*
Measurement C2								
IL-6 [pg/mL]	5.65	0.751 (0.568–0.934)	0.73	0.76	0.75	0.62	0.84	0.017
Measurement: delta C3 vs. C1								
IL-6 [pg/mL]	1.09	0.764 (0.569–0.959)	0.82	0.76	0.78	0.64	0.89	0.018

^2^ ypN+ group: *n* = 12; ypN0 group: *n* = 28.

**Table 3 cancers-16-00757-t003:** Parameters diagnosing groups: TRG1 vs. TRG2 and TRG1 vs. TRG3 ^3^.

	Optimal Threshold	AUC (95% CI)	Sensitivity	Specificity	Accuracy	PPV	NPV	*p*
Measurement: TRG1 vs. TRG2								
IL-6 [pg/mL]	5.16	0.856 (0.674–1.000)	0.80	0.89	0.84	0.89	0.80	0.005
Measurement: TRG1 vs. TRG3								
IL-6 [pg/mL]	6.93	0.796 (0.596–0.997)	0.90	0.69	0.78	0.69	0.90	0.004

^3^ TRG1 group: *n* = 13; TRG2 group: *n* = 12; TRG3 group: *n* = 15.

## Data Availability

The data presented in this study are available on request from the corresponding author.

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
