# Peer review of "Interleukin-6 as a Predictive Factor of Pathological Response to FLOT Regimen Systemic Treatment in Locally Advanced Gastroesophageal Junction or Gastric Cancer Patients"

_cancers, 2024, doi:10.3390/cancers16040757_

Round 1

Reviewer 1 Report

Comments and Suggestions for Authors

The authors provided an interesting evaluation of IL-6 as predictive factor of pathological response in gastric cancer patients receiving FLOT as perioperative treatment.

The topic is worth of interest and I would congratulate the authors for that. However, there are some important lack points that I recommend to improve/add/revise. In particular (in order of appearance in the manuscript):

- I would suggest to add some references in support of the sentences in rows 83-86 (introduction).

- inclusion and exclusion criteria: the authors state that they included patients with ECOG 2; however, no chemotherapy is recommended for those patients and there are no patients with ECOG PS 2 in the table with patients' characteristics. Please clarify this issue.

- I suggest to revise row 136, page 3, erasing the term "we administered" since the schedule that you have described is a non-modified schedule of FLOT;

- The authors used the CTCAE version 4.03, which is not the last version available at the time of the study. Please clarify this issue.

- in the biochemical analysis section, the authors state that patients with NLR >3 received 2 points. However, it is not clear how they identified the cut-off for NLR. Additionally, this description is not related to the results shown (when do you use those scores?). Please clarify;

- I suggest referring to patients who did not have acceded to the centre simply as patients lost (row 234); please revise that sentence.

- The authors state that "we did not find any statistical significance of CEA, CA19.9, CA125, IL-1ß, IL-8, IL-10, F-NLR, LMR, NLR, PLR, CTCs as a predictive biomarker of early response". However, those results are not shown or they did not appear in the manuscript uploaded, while it would be of high interest to see them, also with a comment in the discussion, to prove the authenticity and reproducibility of the data. Please add those data both in the text and as a table;

- the authors state that "Significant difference in the IL-6 serum level before C2 of chemotherapy was recognized comparing TGR1, TGR2 and TGR3 groups (3.76 pg/mL vs 7.07 pg/mL vs 9.43 pg/mL, respectively, p=0.016)". I agree with them that there is a difference; however, how this difference has an impact on survival is not clear since the authors did not show any survival analysis. Please clarify this issue and revise accordingly, eventually by adding a figure in support.

Comments on the Quality of English Language

minor revisions

Author Response

Please find the attachment containing the corrected manuscript.

This paper has been certified by professional language editor.

The authors provided an interesting evaluation of IL-6 as predictive factor of pathological response in gastric cancer patients receiving FLOT as perioperative treatment.

The topic is worth of interest and I would congratulate the authors for that. However, there are some important lack points that I recommend to improve/add/revise. In particular (in order of appearance in the manuscript):

Thank you for your insightful analysis that will help improve the quality of this article.

- I would suggest to add some references in support of the sentences in rows 83-86 (introduction).

We have added references.

- inclusion and exclusion criteria: the authors state that they included patients with ECOG 2; however, no chemotherapy is recommended for those patients and there are no patients with ECOG PS 2 in the table with patients' characteristics. Please clarify this issue.

The Local Bioethics Committee at the Maria Sklodowska-Curie National Research Institute of Oncology (MSCNRIO) in Warsaw approved the protocol version with ECOG 0-1 as inclusion criteria, so only such patients were included in our study. The < character was missing by mistake. It should be ECOG<2. We have corrected it.

- I suggest to revise row 136, page 3, erasing the term "we administered" since the schedule that you have described is a non-modified schedule of FLOT;

We have described the chemotherapy dosage in detail so that there is no doubt that we have used exactly the same dosage method, without any modifications, as in the original Al-Batran et al study. We have corrected it.

- The authors used the CTCAE version 4.03, which is not the last version available at the time of the study. Please clarify this issue.

I have checked the trial documentation, 4.03 was mistake, obviously it should have been version 5.0 (November 27, 2017)

- in the biochemical analysis section, the authors state that patients with NLR >3 received 2 points. However, it is not clear how they identified the cut-off for NLR. Additionally, this description is not related to the results shown (when do you use those scores?). Please clarify;

We used cut-off values for NLR based on previous published data. But in final analysis we did not take under consideration F-NLR because NLR was not statistically significant. We have made comments in the Results section.

- I suggest referring to patients who did not have acceded to the centre simply as patients lost (row 234); please revise that sentence.

We have corrected it.

- The authors state that "we did not find any statistical significance of CEA, CA19.9, CA125, IL-1ß, IL-8, IL-10, F-NLR, LMR, NLR, PLR, CTCs as a predictive biomarker of early response". However, those results are not shown or they did not appear in the manuscript uploaded, while it would be of high interest to see them, also with a comment in the discussion, to prove the authenticity and reproducibility of the data. Please add those data both in the text and as a table;

We have added these results in tables, as well as we have made comments in the Results and Discussion sections. Please find the attachment with the corrected manuscript.

- the authors state that "Significant difference in the IL-6 serum level before C2 of chemotherapy was recognized comparing TGR1, TGR2 and TGR3 groups (3.76 pg/mL vs 7.07 pg/mL vs 9.43 pg/mL, respectively, p=0.016)". I agree with them that there is a difference; however, how this difference has an impact on survival is not clear since the authors did not show any survival analysis. Please clarify this issue and revise accordingly, eventually by adding a figure in support.

We know from previous data, that pathology response has an impact on survival.

We presented the data from our site during Poster Session on WCGIC 2023 in Barcelona (June 2023 Annals of Oncology 34:S134 DOI:10.1016/j.annonc.2023.04.397). According to the Becker classification, TRG1a – 12% (27), TRG1b – 14% (31), TRG2 - 31% (69) and TRG3 - 38% (84) patients were reported, and median overall survival (mOS) was 53.7 (CI 95%; 48.0-59.3), 51.9 (CI 95%; 45.9-57.9), 42.7 (CI 95%; 36.2-49.2), 28.3 (CI 95%; 37.7-44.7), respectively (p < 0.001).

We have added this information to Discussion and References sections.

Reviewer 2 Report

Comments and Suggestions for Authors

The trial seems to be well conducted and the statistical make sense. However, the authors must include and show all outcomes measured in each patient's blood, particularly those related to inflammation. Although the authors mentioned that most of the measured biomarkers did not show significance, a summary table can be very useful to understand the biological variability between cases. Additionally, the authors should consider obesity and visceral fat levels as pro-inflammatory conditions, so their IL6 analysis should at least consider BMI (body mass index) for stratification (that information is not included in the cohort description); however, the authors may find other metrics to include and use in combination to prove the predictive role of IL6 in pCR. The value of IL6 as predictor and its behavior in other human groups requires to understand its level of independence respect to the pro-inflammatory status of each patient, therefore is critical to know if the predictor relies on tumor size, obesity or the intrinsic characteristic of the local population included in this trial

Author Response

Please find the attachment containing the corrected manuscript.

Thank you for your insightful analysis that will help us to improve the quality of this article. We have added information about non statistically significant parameters in the Results as well as in the Discussion sections.

 Thank you for your valuable and accurate comments regarding BMI. It is true that interleukin 6 is a pleiotropic interleukin which plays a great role in inflammation. We also know from numerous reports that it plays a role in neoplastic sarcopenia and sarcopenic obesity. We have added information about BMI to our article and some comments in the Discussion section. We are also in the process of preparing another publication on sarcopenia in this cohort where we will provide information about much better parameters of sarcopenia than BMI, measured in computed tomography before starting treatment and before surgery (HUAC and TPI) and than we will try to correlate with inflammatory parameters.

Please, bear in mind that it was a pilot study aimed at trying to identify only the parameters this which would be worth further investigating on a bigger population. We planned to investigate post-operative histopathological specimens according to tumour size, tumour microenvironment components (CAFs, TAMs, TILs) and to correlate them with IL-6 serum level.

Reviewer 3 Report

Comments and Suggestions for Authors

Gastric cancer, which fortunately has been decreasing in Western countries for about 10 years, has nevertheless found in the diagnostics with preoperative imaging and with the study of histology and microbiology, already carried out in biopsy samples taken during endoscopy, the possibility of neoadjuvant treatment. The FLOT scheme is the most widespread, but there is a second scheme, DOC, which in our opinion is less toxic, but with overlapping results (doi.org/10.1097/CAD.0000000000000877 to be cited in the bibliography). These significantly improve the results of subsequent surgery, sometimes, rarely, with retrostaging. Linitis plastic cases are generally those that are unresponsive. The treatment of the pathology will finally be concluded with adjuvant therapy one month after the operation to improve the outcomes. Follow up and new approaches such as PIPAC and HIPEC are increasingly taken into consideration during antineoplastic treatment. We know that, as in breast cancer, the classic neoplastic antigens CEA and CA19-9 are partially reliable, therefore knowing that interleukin 6 could provide us with reliable results in terms of future prediction of the progress of the disease seems to us to be an absolutely advantageous result for patients and for doctors who follow these pathologies. So the work seems absolutely interesting to us and worthy of being read by a wider audience. However, it is necessary to investigate the large-scale reliability of the results currently obtained with multicentres

​

Comments on the Quality of English Language

the work requires minor revisions of the English language

Author Response

Please find the attachment containing the corrected manuscript.

Thanks for your nice comment. I agree that our results are not currently clinically useful, but they show that it is worth further testing IL-6 in a larger population, especially in multicenter studies.

We presented in another article the results of FLOT in the Polish population - it is not so toxic. (Neoplasma 69(06) DOI:10.4149/neo_2022_220720N734).

We add some information about not statistically significant parameters in Results as well as in Discussion sections.

Round 2

Reviewer 1 Report

Comments and Suggestions for Authors

The authors have nicely addressed all the issues reported.

However, considering the last point that I have raised in my previous report, I would suggest to clarify an issue in the discussion.

In particular, if it is to recognize the authors for their previous study, in which they showed that “Pathological response could be considered as a reliable predictive factor of survival in this setting and could be considered as a valuable surrogate end point of clinical trial apart from OS" in that study population, this is still controversial in the literature. Additionally, as already pointed out, the authors state that "Significant difference in the IL-6 serum level before C2 of chemotherapy was recognized comparing TGR1, TGR2 and TGR3 groups (3.76 pg/mL vs 7.07 pg/mL vs 9.43 pg/mL, respectively, p=0.016)". However, how this difference has an impact on survival in this study population is not clear since the authors did not show any survival analysis, even if they demonstrated the impact of TRG from previous data.

Author Response

Please find attached the article with the requested corrections included.
